# IL-33 Alleviates Postoperative Cognitive Impairment by Inhibiting Hippocampal Inflammation and Upregulating Excitatory Synaptic Number in Aged Mice

**DOI:** 10.3390/brainsci12091244

**Published:** 2022-09-14

**Authors:** Qi Li, Yuqian Zhao, Chuanchuan Shi, Xuemin Song

**Affiliations:** 1Department of Anesthesiology, Zhongnan Hospital of Wuhan University, No. 169 East Lake Road, Wuhan 430071, China; 2Department of Intensive Care Unit, Zhongnan Hospital of Wuhan University, No. 169 East Lake Road, Wuhan 430071, China; 3Department of Intensive Care Unit, Renmin Hospital of Zhengzhou University, No. 7 Wei Wu Road, Zhengzhou 450003, China

**Keywords:** delayed neurocognitive recovery, interleukin-33, inflammation, excitatory synapses

## Abstract

Delayed neurocognitive recovery (dNCR), a postoperative complication that occurs in elderly patients, still lacks effective treatment. Interleukin-33 (IL-33) has been proved to modulate neuroinflammation and synaptic plasticity, among other effects, but the role of IL-33 in dNCR is not clear. We established a dNCR model in aged mice by laparotomy under sevoflurane anesthesia. Cognition was evaluated by Morris water maze (MWM) and fear conditioning test (FCT). Immunofluorescence was used to detect the density of IL-33 and glial fibrillary acidic protein (GFAP) co-localization, ionized calcium-binding adapter molecule 1, vesicular glutamate transporter 1 (vGlut1) and postsynaptic density protein-95 (PSD95) co-localization in the hippocampus. IL-33, GFAP, vGlut1 and PSD95 were tested by Western blotting. Enzyme-linked immunosorbent assay was used to detect the levels of tumor necrosis factor-alpha (TNF-α), interleukin-1 beta (IL-1β) and IL-10. Surgery/anesthesia reduced the level of IL-33 in the hippocampus. Intraperitoneal injection of 200 ng IL-33 per mouse significantly decreased the latency to the platform and increased the number of platform crossings and the target quadrant dwell time in MWM, while increasing the freezing time in the context test of FCT. Furthermore, IL-33 inhibited microglial activation and the release of TNF-α and IL-1β while upregulating the markers of excitatory synapses vGlut1 and PSD95. Our findings indicated that IL-33 improved cognition by inhibiting the hippocampal inflammatory response and upregulating the number of excitatory synapses. Therefore, IL-33 is a potential drug for the treatment of dNCR.

## 1. Introduction

Delayed neurocognitive recovery (dNCR), a widely known postoperative central nervous system (CNS) complication, frequently occurs in older surgical patients [1]. dNCR is characterized by impairments in learning, memory, attention and executive functions occurring within 30 days of surgery [2]. The occurrence of dNCR prolongs the hospital stay and reduces quality of life for patients while imposing a serious economic burden on society [3]. Although various mechanisms such as CNS inflammation, mitochondrial damage, blood-brain barrier damage and oxidative stress have been proved to be related to the occurrence of dNCR [4], there are still no effective therapeutic measures.

Interleukin-33 (IL-33), an alarmin of the IL-1 family, is a key mediator of innate immune responses [5]. Numerous studies have shown that IL-33 is mainly secreted by astrocytes and microglia but not neurons [6,7]. However, other studies indicated that IL-33 is also expressed in neurons and oligodendrocytes [8,9]. IL-33 participates in the signaling pathway by binding to the heterodimeric receptor of the complex composed of suppression of tumorigenicity 2 (ST2) and IL-1R accessory protein (IL-1RAP). IL-33 is involved in intracellular biological responses by recruiting proteins such as myeloid differentiation factor 88 (MyD88) and IL-1R-associated kinase 1 (IRAK1) to activate a variety of intracellular signaling molecules such as nuclear factor-kappa B (NF-κB) and mitogen-activated protein kinase [10,11]. IL-33 inhibited hippocampal inflammation by blocking the activation of NF-κB, eventually improving spatial learning and memory ability in recurrent neonatal seizure (RNS) rats [12]. In addition, Wang et al. found that inhibition of hippocampal CA1 neuronal activity increased the release of astrocyte-derived IL-33, thereby increasing the number of excitatory synapses [13]. Conversely, there are also some reports on the pro-inflammatory effects of IL-33 in the CNS, such as the lipopolysaccharide (LPS)-induced CNS inflammation mouse model [14,15]. The above-mentioned studies indicate that IL-33 may play opposite roles in different CNS inflammation-related diseases. The role and potential mechanisms of IL-33 in dNCR are not clear. Aged mice were often used to construct dNCR animal models [16]; therefore, we established an aged mice dNCR model to explore the role of IL-33 in the present study.

## 2. Methods

### 2.1. Animals and Groups

A total of fifty 18-month-old male C57BL/6 mice (30–40 g) were purchased from Tianqin Biotechnology Company Limited (Changsha, China). Mice were fed ad libitum in a 12 h light/dark cycle environment (humidity was 50% ± 5%, and the temperature was 22 ± 0.5 °C) and acclimatized to the raising conditions for 14 days before the study. The study was approved by the Animal Ethics Committee of Wuhan University (ethics number: WQ20210298), and all experiments were performed in accordance with the Guiding Principles for the Care and Use of Animals in Research, the ARRIVE 2.0 guidelines [17]. Mice with abnormal motility and somatic diseases were excluded, and two mice were excluded due to abdominal tumors in the present study.

To investigate the effect of surgery/anesthesia on the levels of IL-33 in the hippocampus of mice, we divided the mice into control and surgery groups (4 mice per group) in the first step of the experiment. The second part of the experiment was used to investigate the effect of IL-33 on cognitive function and its mechanism. Mice were divided into control, IL-33, surgery and surgery + IL-33 groups (10 mice per group). The diagram of the study is shown in Figure 1A.

### 2.2. dNCR Mice Model Drug Administration

The dNCR mouse model was constructed based on a previous report [16]. Briefly, mice were induced with 5% sevoflurane for 3 min, followed by surgery under 2.0–2.5% sevoflurane, and maintained body temperature on a heating plate. After shaving and sterilizing the surgical site, a 1.5 cm longitudinal incision was performed 5 mm below the right rib, and subsequently, the abdominal organs, including the liver, spleen, kidneys and bowel, were explored. Next, 10 cm of small intestine was vigorously rubbed with the index finger and thumb for 0.5 min. After the small intestine was put back, the wound was closed with 4-0 sutures. At the end of surgery/anesthesia, lidocaine cream was used to alleviate incision pain every 8 h for 3 days after surgery. Mice in the control group only received shaving and sterilization.

Murine recombinant IL-33 was purchased from Biolegend (580506), and 200 ng per mouse [18] was injected intraperitoneally from the day of surgery to 3 days postoperatively.

### 2.3. Morris Water Maze Test (MWM)

The MWM was performed to detect hippocampus-dependent learning and memory in rodents [19]. The test contains two parts: place navigation and spatial probe testing. In the place navigation test, the swimming speed and the latency to find the platform of the mice were recorded. Removing the platform on the third postoperative day for spatial probe testing, the number of platform crossings and the target quadrant dwell time within 60 s were recorded. An ANY-maze video tracking system was applied to record the movement trajectory of mice.

### 2.4. Fear Conditioning Test (FCT)

FCT consists of a training part to build long-term memory and a testing part [20]. In the training part, mice were acclimated to the chamber for 2 min and given a sound stimulus of 70 dB (conditional stimulus). After a 25 s interval, an electric foot shock (unconditional stimulus) was performed (0.7 mA, 2 s). After two pairs of conditional-unconditional stimuli were performed, the mice stayed in the chamber for 1 min. The testing part contained context and tone tests. In the context test, the mice were allowed to explore freely in the chamber for 5 min without any stimulation. In the tone test (2 h after the context test), the environment of the chamber was changed. Mice were given 70 dB sound stimulation (total 180 s) without a foot shock. The freezing time was recorded by the ANY-maze tracking system.

### 2.5. Western Blotting

The mouse hippocampus was lysed by radioimmunoprecipitation lysis buffer (Solarbio, Beijing, China, R0010) with a protease phosphatase inhibitor mixture (Solarbio, P1260). A BCA kit (Solarbio, PC0020) was applied to quantify the total protein concentration. Protein samples (40 μg per lane) were separated by 10% SDS-PAGE (Solarbio, P1200) and transferred onto polyvinylidene fluoride (PVDF) membranes (Millipore, MA, USA, IPFL00010). The membranes were blocked with 5% skim milk (Solarbio, D8340) at room temperature (1 h) and subsequently immunoblotted with primary antibodies overnight at 4 °C: IL-33 (R&D, MND, USA, 1:500, AF3626), glial fibrillary acidic protein (GFAP) (Abcam, Cambridge, UK, 1:1000, ab7260), vesicular glutamate transporter 1 (vGlut1) (Abcam, 1:1000, ab227805), postsynaptic density protein-95 (PSD95) (Abcam, 1:1000, ab13552) and glyceraldehyde-3-phosphate dehydrogenase (GAPDH) (Abcam, 1:2000, ab8245). Next, 1X tris-buffered saline Tween (TBST) (Solarbio, T1081) was used to wash the membranes 3 times. They were then incubated with a secondary antibody at room temperature (40 min). Immunoreactivity was detected using enhanced chemiluminescence (Solarbio, P0018), and bands were measured using Image J analysis software (Version 1.50i, Rockville, MD, USA).

### 2.6. Enzyme-Linked Immunosorbent Assay (ELISA)

The levels of tumor necrosis factor-alpha (TNF-α) (Boster Biological Technology, Wuhan, China, EK0527), interleukin-1 beta (IL-1β) (Boster, EK0394) and IL-10 (Boster, EK0417) in the hippocampus were detected by an ELISA kit in accordance with the manufacturer’s instructions and represented as pg/mL.

### 2.7. Immunofluorescence Staining

The mice were transcardially perfused with ice-cold phosphate-buffered saline (PBS) followed by 4% paraformaldehyde under isoflurane deeply anesthetized. The brain was collected in 4% formaldehyde for 24 h at 4 °C, followed by 30% sucrose dehydration for 48 h at 4 °C. Coronal 30 μm-thick sections were cut on a cryostat (CM1900; Leica, Wiesbaden, Germany). The sections were washed 3 times with PBS and blocked with 5% goat serum at room temperature for 1 h. Then, the sections were incubated with primary antibodies overnight at 4 °C: IL-33 (R&D, 1:200, AF3626), GFAP (Abcam, 1:500, ab7260), vGlut1 (Abcam, 1:100, ab227805), PSD95 (Abcam, 1:100, ab13552) and ionized calcium-binding adapter molecule 1 (Iba1) (Wako, 1:500, 019-19741). The sections were incubated with secondary antibodies (Alexa Fluor 488 and 594) for 1 h at 37 °C after washing with PBS 3 times, followed by DAPI staining for 8 min at room temperature. Then, the sections were washed with PBS 3 times and mounted with 70% glycerol.

### 2.8. Statistical Analysis

The statistical analysis of the data was performed by GraphPad Prism 8 (GraphPad, New York, NY, USA). Quantitative data were expressed as the mean ± standard deviation (SD), and the Shapiro–Wilk test showed that the data were normally distributed. Analysis of variance (ANOVA) followed by Bonferroni’s post hoc test and two-tailed unpaired *t*-test were used to test for statistical significance. A *p*-value less than 0.05 was considered statistically significant.

## 3. Results

### 3.1. Surgery/Anesthesia Decreases the Hippocampal Astrocyte-Derived IL-33

To identify the effect of surgery/anesthesia on the expression of IL-33, we measured total and astrocyte-derived IL-33 protein levels in the hippocampus. The data showed that surgery/anesthesia decreased the astrocyte-derived IL-33 protein levels (Figure 1B,C *p* < 0.05) compared with the control group. Compared with the control group, surgery/anesthesia decreased the total levels of IL-33 (Figure 1D,F *p* < 0.01) and increased the levels of GFAP (Figure 1D,F *p* < 0.05) in the hippocampus. These results indicated that surgery/anesthesia decreased the astrocyte-derived IL-33, which may be associated with cognitive impairment.

### 3.2. IL-33 Alleviates Surgery/Anesthesia-Induced Cognitive Impairment in Mice

MWM was used to evaluate spatial learning and memory [19]. In the MWM, there was no significant difference in preoperative latency (Figure 2A *p* > 0.05) and swimming speed (Figure 2B *p* > 0.05) among the four groups. Furthermore, both surgery/anesthesia and IL-33 did not affect the swimming speed (Figure 2C *p* > 0.05) of mice 3 days postoperatively. However, surgery/anesthesia increased the latency to platform (Figure 2D *p* < 0.01) and decreased the times of platform crossing (Figure 2E *p* < 0.01) and time spent in the target quadrant (Figure 2F *p* < 0.05) of the mice, while IL-33 decreased the latency to platform (Figure 2D *p* < 0.05) and increased the times of platform crossing (Figure 2E *p* < 0.05).

Hippocampus-dependent memory was detected by a context test, while hippocampus-independent memory in rodents was detected by a context test in FCT [21]. Surgery/anesthesia significantly reduced the freezing time of mice in the context test (Figure 2G, *p* < 0.01) at 3 days postoperatively but not in the tone test (Figure 2H, *p* > 0.05), which was consistent with previous studies [16]. In addition, IL-33 increased the freezing time of mice in the context test (Figure 2G, *p* < 0.05). These data indicated that IL-33 could alleviate surgery/anesthesia-induced hippocampus-independent memory impairment in added mice.

### 3.3. IL-33 Inhibits Surgery/Anesthesia-Induced Hippocampal Inflammation in Aged Mice

Iba1, as a microglia marker, has usually been used to reflect CNS inflammation [22]. The results indicated that surgery/anesthesia increased the fluorescence intensity of Iba1 (Figure 3A,B, *p* < 0.001) in the hippocampus of mice, while IL-33 decreased it (Figure 3A,B, *p* < 0.01). Moreover, surgery/anesthesia increased the levels of TNF-α (Figure 3C, *p* < 0.01) and IL-1β (Figure 3D, *p* < 0.001) and decreased the level of IL-10 (Figure 3E, *p* < 0.01), while IL-33 decreased TNF-α (Figure 3C, *p* < 0.05) and IL-1β (Figure 3D, *p* < 0.01) but not IL-10 (Figure 3E, *p* > 0.05). These results demonstrated that IL-33 may alleviate surgery/anesthesia-induced hippocampus-independent memory impairment via inhibiting hippocampal inflammation.

### 3.4. IL-33 Upregulates the Hippocampal Number of Excitatory Synapses in dNCR Aged Mice

Rodents’ hippocampal excitatory synaptic density correlates with cognitive function, and co-localization of vGlut1 and PSD95 is commonly used to label excitatory synaptic neurons [23]. We found that surgery/anesthesia decreased the density of excitatory synapses (Figure 4A–E, *p* < 0.05), whereas IL-33 reversed the reduction in glutamatergic synapse density (Figure 4A–E, *p* < 0.05) in the hippocampus of dNCR aged mice. These data indicated that IL-33 may improve surgery/anesthesia-induced hippocampus-independent memory impairment via upregulating the number of excitatory synapses in the hippocampus.

## 4. Discussion

IL-33 plays a key role in learning and memory via regulating hippocampal inflammation and synaptic plasticity [24], but the underlying molecular and cellular mechanisms in dNCR remain unclear. Our study showed that surgery/anesthesia induced postoperative cognitive impairment accompanied by decreased astrocyte-derived IL-33 in aged mice. Intraperitoneal injection of IL-33 could improve the cognitive function of dNCR mice while inhibiting hippocampal inflammation and rescuing excitatory synaptic numbers. Hence, these findings demonstrated that exogenous IL-33 supplementation is effective in improving the cognitive function of dNCR mice, which is related to the role of IL-33 in anti-inflammatory and upregulation of excitatory synaptic number. As far as we know, the present study is the first to report the role of IL-33 in dNCR.

dNCR can complicate surgery and increase morbidity and mortality, especially in the elderly [3]. dNCR is characterized by memory impairment, decreased attention and a decline in information handling [2]. Although studies on dNCR are in full swing, its pathogenesis remains unclear. dNCR shares a similar pathogenesis with psychiatric disorders, such as Alzheimer’s disease (AD), dementia and depression [4]. Researchers have proposed various pathophysiological mechanisms for dNCR, including neuroinflammation [25], synaptic dysfunction [16], oxidative stress [26], etc. Since neuroinflammation is the most widely studied pathogenesis of dNCR and the synaptic number is directly related to cognitive function, we explored the role of neuroinflammation and synaptic number in dCNR in the present study.

IL-33 is highly expressed in the brain, especially in astrocytes [27]. As research on IL-33 has progressed, it has been shown to be expressed in microglia, neurons and oligodendrocytes [18]. Moreover, IL-33 could also be released from dead and injured cells [28]. Hence, we detected the total and astrocyte-derived IL-33 in the hippocampus. The data showed that surgery/anesthesia significantly decreased the total as well as astrocyte-derived IL-33 in the hippocampus at 3 days postoperatively. A clinical study has shown that IL-33 was significantly reduced in cerebrospinal fluid and serum of AD and mild cognitive impairment patients [29]. IL-33 administration could promote the polarization of microglia toward the anti-inflammatory type and decrease soluble β-amyloid levels and amyloid plaque deposition in AD mice [18]. The above-mentioned studies are consistent with our findings and indicate that IL-33 plays an important role in cognition.

In the present study, 200 ng IL-33 per mouse [18] was selected to investigate its effect on cognitive function in dNCR mice. We found that IL-33 could significantly decrease the latency to the platform and increase the times of platform crossing and time spent in the target quadrant of dNCR mice 3 days postoperatively. In the FCT test, IL-33 increased the freezing time in the context test compared with dNCR mice. These findings demonstrated that an intraperitoneal injection of 200 ng IL-33 could improve hippocampus-dependent learning and memory of dNCR mice but did not affect aged mice in the control group. Surgery/anesthesia did not affect the freezing time of dNCR mice in the tone test, which is consistent with previous studies [16,20] and indicated that surgery/anesthesia did not impair hippocampus-independent memory. In AD mouse [18,30] and RNS rat [12] models, IL-33 has been proven to improve cognitive defects via anti-inflammation. Conversely, bilateral intrahippocampal injection of IL-33 (400 ng) induced IL-1β overexpression by microglia cells and cognitive impairment in mice [31]. IL-33 deficiency relieved LPS-induced inflammatory response in mice and BV2 microglia activation [14]. Hence, IL-33 may have opposite effects on cognitive function via divergent regulation of inflammation in the CNS and may exert neuroprotective effects only at appropriate doses.

IL-33 has been shown to affect cognitive function by modulating neuroinflammation, synaptic plasticity, apoptosis and autophagy [24]. Numerous studies have shown that the overactivation of microglia and the release of inflammatory factors (e.g., TNF-α, IL-1β, IL-6 and IL-10) induce the occurrence of dNCR by impairing neuronal function [20,32]. In the present study, we found that IL-33 significantly inhibited microglia activation and the release of inflammatory cytokines (TNF-α and IL-1β). These data indicated that IL-33 may improve the cognition of dNCR mice by inhibiting hippocampal inflammatory response, which was consistent with the role of IL-33 in AD and RNS [12,18]. ST2/IL-1RAP complex, as a receptor for IL-33, has been verified to be expressed on microglia. Therefore, we speculated that IL-33 exerts its anti-inflammatory effects through the ST2/IL-1RAP receptor complex on microglia. IL-10, as an anti-inflammatory factor and the marker of M2 microglia, has been verified to be involved in the cognition of dNCR [33]. We found that surgery/anesthesia decreased the level of IL-10, which was in line with the above-mentioned study. However, IL-33 could not reverse the decrease in IL-10, suggesting that the effects of IL-33 on inflammatory factors may be targeted.

Cognitive function is associated with synaptic plasticity, including the number and function of synapses [16,23,34]. Previous studies have shown that a decrease in the number of hippocampal excitatory synapses is associated with cognitive decline in mice [16,23]. IL-33 administration reversed the long-term potentiation (LTP) impairment and contextual memory deficits in AD mice [35] and promoted the formation of functional excitatory synapses in hippocampal CA1 neurons [13]. Hence, we hypothesized that the effects of IL-33 on cognitive function might be related to the effects on excitatory synapses. We used the co-localization of vGlut1 (a presynaptic marker of excitatory neurons) and PSD95 (a postsynaptic marker) to indicate the number of excitatory synapses, and our results indicated that IL-33 could promote the formation of excitatory synapses, which was in line with previous studies.

The current limitations of this study are as follows. First, although previous studies have demonstrated that IL-33 is predominantly expressed in cells, as well as the fact that we also found a decrease in astrocyte-derived IL-33, we cannot exclude the role of IL-33 in other cell types. Second, we chose 200 ng IL-33 per mouse according to the previous study [18] and did not administer a multi-dose of IL-33, which may have missed the effects of other doses on dNCR mice. Third, we only detected the number of excitatory synapses, but not LTP and other indicators of synaptic function.

## 5. Conclusions

In the present study, surgery/anesthesia reduced the levels of IL-33 in the hippocampus of dNCR mice. IL-33 significantly inhibited the microglia activation and the release of inflammatory factors (TNF-α and IL-1β) and increased the number of excitatory synapses, ultimately improving the cognition of dNCR mice. Our study not only elucidates the mechanism by which IL-33 improves cognition but also provides a theoretical basis for IL-33 as a potential therapeutic agent for dNCR.

## Figures and Tables

**Figure 1 brainsci-12-01244-f001:**
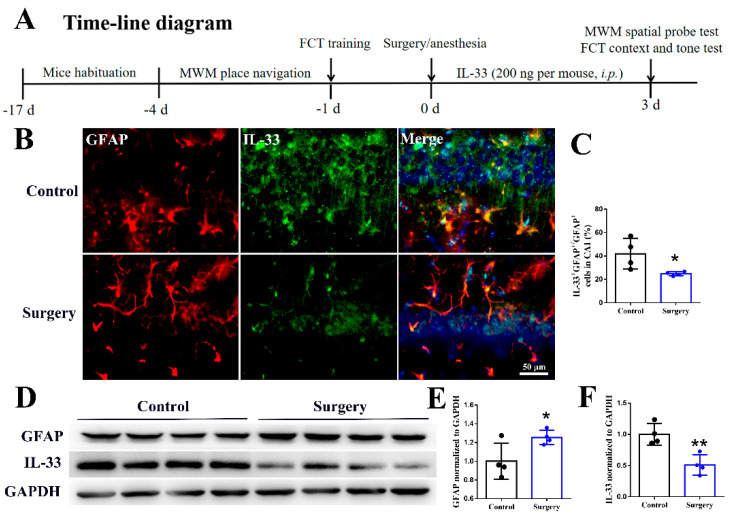
Surgery/anesthesia decreases the hippocampal astrocyte-derived IL-33. Timeline of the study (**A**). The levels of IL-33 and GFAP were detected by immunofluorescence (**B**,**C**) and Western blot (**D**–**F**), respectively 3 days after surgery. Data are expressed as the mean ± SD (two-tailed unpaired *t*-test, n = 4 per group). * *p* < 0.05 and ** *p* < 0.01 versus the control group.

**Figure 2 brainsci-12-01244-f002:**
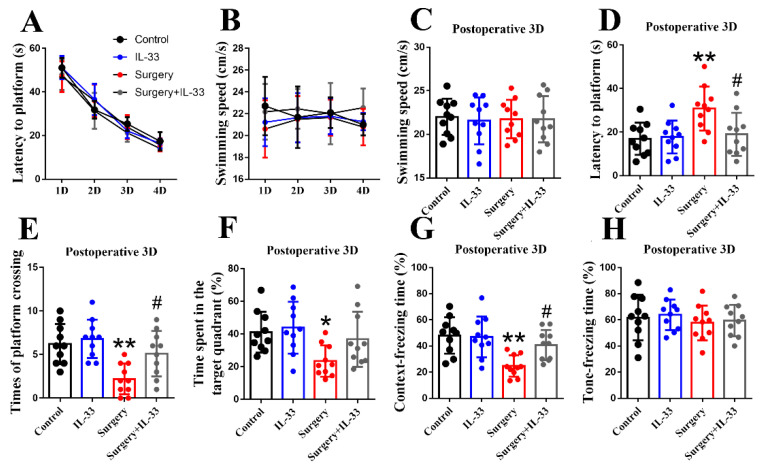
IL-33 alleviates surgery/anesthesia-induced cognitive impairment in mice. Latency to platform (**A**) and swimming speed (**B**) were recorded in the place navigation of MWM on days 1–4 before surgery/anesthesia. Swimming speed (**C**), latency to platform (**D**), times of platform crossing (**E**), the time spent in the target quadrant (**F**), freezing time in the context test (**G**) and tone test (**H**) were recorded on days 3 after surgery. Data are expressed as the mean ± SD (repeated measure two-way ANOVA followed by Bonferroni’s post hoc test, n = 10 per group). * *p* < 0.05 and ** *p* < 0.01 versus the control group; # *p* < 0.05 versus the surgery group.

**Figure 3 brainsci-12-01244-f003:**
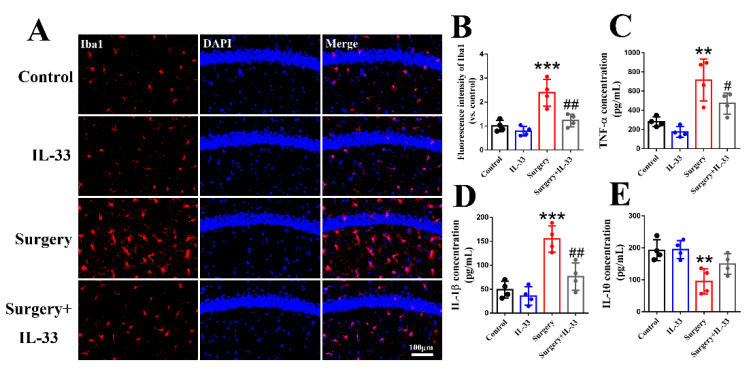
IL-33 inhibits surgery/anesthesia-induced hippocampal inflammation in aged mice. Activation of microglia in the hippocampus was indicated by Iba1 (**A**,**B**). The levels of TNF-α (**C**), IL-1β (**D**) and IL-10 (**E**) were detected by ELISA 3 days after surgery. Data are expressed as the mean ± SD (one-way ANOVA followed by Bonferroni’s post hoc test, n = 4 per group). ** *p* < 0.01 and *** *p* < 0.001 versus the control group; # *p* < 0.05 and ## *p* < 0.01 versus the surgery group.

**Figure 4 brainsci-12-01244-f004:**
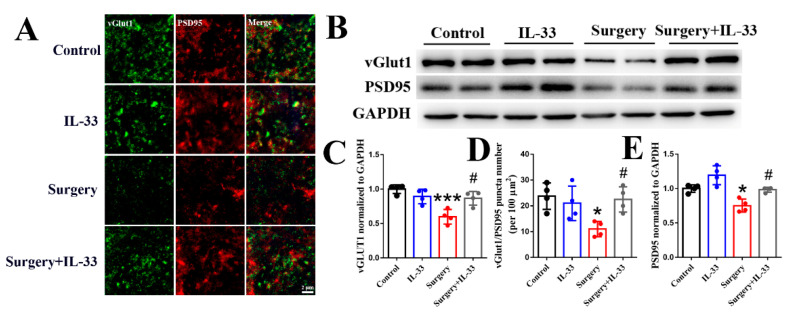
IL-33 upregulates the hippocampal number of excitatory synapses in dNCR aged mice. The levels of vGlut1 and PSD95 were detected by immunofluorescence (**A**,**C**) and Western blot (**B**,**D**,**E**), respectively, 3 days after surgery. Data are expressed as the mean ± SD (one-way ANOVA followed by Bonferroni’s post hoc test, n = 4 per group). * *p* < 0.05 and *** *p* < 0.001 versus the control group; # *p* < 0.05 versus the surgery group.

## Data Availability

All data supporting the findings of this study are available to the corresponding author upon request.

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
