# Peer review of "IL-33 Alleviates Postoperative Cognitive Impairment by Inhibiting Hippocampal Inflammation and Upregulating Excitatory Synaptic Number in Aged Mice"

_brainsci, 2022, doi:10.3390/brainsci12091244_

Round 1

Reviewer 1 Report

Dear Authors, many thanks for the opportunity to read this awesome research

What I really can say?

This research is an interesting prospective about delirium in experimental mice model.

This paper is ver novel, well written and well done Really I do not have any concerns References are appropriate  The study design is very well done This type of research is not easy to perform The authors performed a great research

The Introduction is concise, reports a novelty topic

The methods are well described

Results: nice graphs, nice photos, results are clearly expressed 

Discussion is appropriate 

Conclusion : correct

Well done

Author Response

Thanks for your comments

Reviewer 2 Report

Thank you for the opportunity to review the work.

The work is very innovative, interesting and I believe that the authors should continue their research on a similar subject.

Here are some questions:

- Why did the test group consist only of males?

- Please enter the number of mice in subsection 2.1.

- Were there any exclusion criteria?

- Were all mice purchased eligible for the study? If not, what were the exclusion criteria?

- There is no clear list of inclusion and exclusion criteria for the study

Despite the comments I added, I believe that the work is very interesting, well written and should be published with minor corrections.

Good luck

Reviewer

Author Response

1. Why did the test group consist only of males?

Response: That is a good question. To exclude the effect of estrogen on the experimental results, we preferred male mice for the experiments. Although a recent report demonstrated that there is no difference in behaviors and physiological state between female and male mice (Shansky. Science, 2019). Therefore, only male mice were included in the present study.

Reference:

Shansk, R, M. Are hormones a “female problem” for animal research? Science. 2019.

2. Please enter the number of mice in subsection 2.1.

Response: The number of mice (total 50 mice) was added in the revised manuscript (Method-2.1 animal, line 62).

3. Were there any exclusion criteria?

Response: According to previous studies, no exclusion criteria for dNCR mice model construction were reported. These 50 mice were not significantly different in either weight or appearance. Therefore, no mouse was excluded before surgery.

4. Were all mice purchased eligible for the study? If not, what were the exclusion criteria?

Response: Thanks for your insightful comments. First, two mice were found to have tumors in their abdominal cavity during surgery and were euthanized. Second, water maze results suggest no significant difference in motor ability in mice either before or after surgery, and we did not observe any abnormalities in the locomotor ability of any of the mice. Therefore, total two mice were excluded in the present study.

5. There is no clear list of inclusion and exclusion criteria for the study

Response: Thanks for your suggestions. The exclusion criteria were added in the revised manuscript (Method-2.1 animal, line 69-70)- “Mice with abnormal motility and somatic diseases were excluded, and two mice were excluded due to abdominal tumors in the present study.”

Reviewer 3 Report

The authors presented the work to investigate the role of Interleukin-33 (IL-33) in Delayed neurocognitive recovery (dNCR). The work is well written and overall English is good. 

However, I have some questions:

Line 56 The abovementioned studies suggest that IL-33 may play opposite roles in different animal models. 

The term different animal models are unclear. Can you be more specific? Is it always related to the models of mice or other animals?

Line 58 In the present study, we established an aged mice dNCR model to explore the role of IL-33.

The aim is clear, however, it leaves a question of why on mice. I suggest adding some sentences before why the study is performed on mice and/or they are related studies. 

Methods

What I didn't find in the text is how many mice were used for the control group and how many for the experimental group.

In general, I miss the information about the control group. There is a lack of information on procedures that were done on mice selected to be in the control group.

Line 200 To be consistent I suggest changing "vGLUT1" to "vGlut1".

I am suggestion also to add the weak points of the work. For me the cause of dNCR and inhibition of hippocampal inflammation and other parameters measured in the paper are hypothetical, and in my opinion, they should be addressed in the discussion.

Author Response

1. Line 56 The abovementioned studies suggest that IL-33 may play opposite roles in different animal models. The term different animal models are unclear. Can you be more specific? Is it always related to the models of mice or other animals?

Response: We apologized for this unclear description. According to previous studies, IL-33 plays an anti-inflammatory role in AD mouse model (Fu, et al. Proc Natl Acad Sci U S A. 2016) and recurrent neonatal seizure (RNS) rat model (Gao, et al. Front Mol Neurosci. 2017), and a pro-inflammatory role in lipopolysaccharide-induced CNS inflammation model (Cao, et al. J Neuroinflammation. 2018). Therefore, IL-33 may play opposite roles in different CNS inflammation-related diseases (Introduction-line 56-59).

References:

Fu, A.K., Hung, K.W., Yuen, M.Y., Zhou, X., Mak, D.S., Chan, I.C., Cheung, T.H., Zhang, B., Fu, W.Y., Liew, F.Y., et al. IL-33 ameliorates Alzheimer's disease-like pathology and cognitive decline. Proc Natl Acad Sci U S A. 2016, 113, E2705-E2713.

Gao, Y., Luo, C.L., Li, L.L., Ye, G.H., Gao, C., Wang, H.C., Huang, W.W., Wang, T., Wang, Z.F., Ni, H., et al. IL-33 Provides Neuroprotection through Suppressing Apoptotic, Autophagic and NF-κB-Mediated Inflammatory Pathways in a Rat Model of Recurrent Neonatal Seizure. Front Mol Neurosci. 2017, 10, 423

Cao, K.; Liao, X.; Lu, J.; Yao, S.; Wu, F.; Zhu, X.; Shi, D.; Wen, S.; Liu, L.; Zhou, H. IL-33/ST2 plays a critical role in endothelial cell activation and microglia-mediated neuroinflammation modulation. J Neuroinflammation. 2018, 15, 136.

2. Line 58 In the present study, we established an aged mice dNCR model to explore the role of IL-33. The aim is clear, however, it leaves a question of why on mice. I suggest adding some sentences before why the study is performed on mice and/or they are related studies.

Response: Thanks for your suggestion. The sentence “Aged mice were often used to construct dNCR animal models, therefore we established an aged mice dNCR model to explore the role of IL-33 in the present study.” Has been added in the revised manuscript. (Introduction-line 60-62)

Methods

3. What I didn't find in the text is how many mice were used for the control group and how many for the experimental group.

Response: The number of mice was shown in the figure legends. Now, the groups and number of mice was added in the revised manuscript. (Method-2.1 animals and groups-line 74-78).

4. In general, I miss the information about the control group. There is a lack of information on procedures that were done on mice selected to be in the control group.

Response: The sentence “Mice in the control group only received shaving and sterilization” was added in the revised manuscript. (Method-2.2 dNCR mice model drug administration-line 90-91)

5. Line 200 To be consistent I suggest changing "vGLUT1" to "vGlut1".

Response: "vGLUT1" has been corrected to "vGlut1".

6. I am suggestion also to add the weak points of the work. For me the cause of dNCR and inhibition of hippocampal inflammation and other parameters measured in the paper are hypothetical, and in my opinion, they should be addressed in the discussion

Response: Thanks for your comments, and the discussion has been modified in the revised manuscript.

Round 2

Reviewer 3 Report

The Authors have addressed all of my concerns with the original manuscript. The revised manuscript is ready for publication